# Effectiveness of Two Models of Telerehabilitation in Improving Recovery from Subacute Upper Limb Disability after Stroke: Robotic vs. Non-Robotic

**DOI:** 10.3390/brainsci14090941

**Published:** 2024-09-21

**Authors:** Arianna Pavan, Alessio Fasano, Stefania Lattanzi, Laura Cortellini, Valeria Cipollini, Sabina Insalaco, Maria Cristina Mauro, Marco Germanotta, Irene Giovanna Aprile

**Affiliations:** Neuromotor Rehabilitation Department, IRCCS Fondazione Don Carlo Gnocchi ONLUS, 50143 Florence, Italy; apavan@dongnocchi.it (A.P.); slattanzi@dongnocchi.it (S.L.); lcortellini@dongnocchi.it (L.C.); vcipollini@dongnocchi.it (V.C.); sinsalaco@dongnocchi.it (S.I.); mmauro@dongnocchi.it (M.C.M.); mgermanotta@dongnocchi.it (M.G.); iaprile@dongnocchi.it (I.G.A.)

**Keywords:** stroke, rehabilitation, telehealth, home-based, upper extremity, robotics, robot-assisted rehabilitation training, virtual reality, rehabilitation protocol

## Abstract

Background/Objectives: Finding innovative digital solutions is fundamental to ensure prompt and continuous care for patients with chronic neurological disorders, whose demand for rehabilitation also in home-based settings is steadily increasing. The aim is to verify the safety and the effectiveness of two telerehabilitation (TR) models in improving recovery from subacute upper limb (UL) disability after stroke, with and without a robotic device. Methods: One hundred nineteen subjects with subacute post-stroke UL disability were assessed for eligibility. Of them, 30 patients were enrolled in the study and randomly assigned to either the Robotic Group (RG), undergoing a 20-session TR program, using a robotic device, or the Non-Robotic Group (NRG), undergoing a 20-session TR program without robotics. Clinical evaluations were measured at baseline (T0) and post-intervention (T1, 5 weeks after baseline), and included assessments of quality of life, motor skills, and clinical/functional status. The primary outcome measure was the World Health Organization Disability Assessment Schedule 2.0, evaluating the change in perceived disability. Results: Statistical analysis shows that patients of both groups improved significantly over time in all domains analyzed (mean decrease from baseline in the WHODAS 2.0 of 6.09 ± 2.62% for the NRG, and of 0.76 ± 2.21% for the RG), with a greater improvement of patients in the NRG in motor (Fugl-Meyer Assessment Upper Extremity—motor function, Box and Block Test) and cognitive skills (Trail Making Test-A). Conclusions: This study highlights the potential of TR programs to transform stroke rehabilitation by enhancing accessibility and patient-centered care, promoting autonomy, improving adherence, and leading to better outcomes and quality of life for stroke survivors.

## 1. Introduction

The progressive aging of the population has a major impact on the development of health policy, particularly as longer life expectancy is associated with a higher likelihood of chronic disease for a given individual [1]. In Italy, for instance, over 3 million people, representing approximately 5% of the population, experience cognitive and/or motor disabilities that significantly limit their daily activities and require ongoing care and assistance [2].

Chronic neurological diseases are a leading cause of disability in daily life, and they rank as the second-leading cause of death globally [3]. Among these neurological conditions, stroke is particularly notable for its significant impact on patients’ quality of life, contributing to the highest number of years lived with disability, according to the World Health Organization [4].

Rehabilitation is widely recognized as the preferred treatment for managing chronic neurological conditions. A recent review highlighted that in Europe alone, over 300 million people require rehabilitation services, with approximately one-fifth needing specialized neurorehabilitation [5]. Despite the growing demand, current rehabilitation services are often highly specialized, expensive, and predominantly hospital-based, limiting access for many patients. Effective rehabilitation requires early and intensive intervention, underscoring the urgency for health systems to develop viable and sustainable solutions beyond the traditional hospital setting.

Robotic rehabilitation for individuals with subacute upper limb (UL) disability following a stroke is increasingly endorsed as an adjunct to conventional therapy in select stroke guidelines [6]. These robotic systems are viewed as a promising addition, offering repetitive and precise movements that may enhance recovery.

However, despite growing interest, scientific evidence comparing the efficacy of robotic rehabilitation to traditional approaches remains inconclusive. Some studies show improvements in motor function and faster recovery with robotics, while others report minimal differences [7,8,9,10]. The variability in patient responses and rehabilitation protocols complicates drawing definitive conclusions [11,12].

Thus, while robotic rehabilitation represents an exciting advancement, its full potential compared to conventional therapy is still under investigation.

Robotic systems have proven effective in supporting the recovery of motor and cognitive function in stroke patients [7]. Large randomized clinical trials have demonstrated their equivalence to traditional rehabilitation when a similar treatment dose is provided [7,13]. Additionally, these technologies enable the intensification of treatment and the development of personalized treatment plans based on objective and measurable outcomes, especially in improving the recovery from subacute UL disability after stroke [14,15,16,17,18,19].

Furthermore, the use of health technologies, particularly those developed and validated through an evidence-based approach, and the use of digital platforms to deliver remote rehabilitation services, represent a new frontier in the integrated and continuous care of chronic neurological disabilities [20].

This approach aims to provide patients, their families, and healthcare professionals with rigorously validated tools that are both accessible and user-friendly [21].

The COVID-19 pandemic presented unprecedented challenges for rehabilitation services, particularly in terms of the ongoing need for patient support and the security of access to healthcare facilities [22]. This situation accelerated the shift to telemedicine and emphasized the importance of providing effective rehabilitation tools in safe environments, including patients’ homes [23,24,25], even in non-emergency situations. This shift highlighted the critical need to develop and implement home rehabilitation protocols that can deliver care remotely while ensuring effectiveness and safety [26,27,28]. In response to these challenges, the current study aims to evaluate the safety and effectiveness of a non-robotic TR program versus a TR program using a robotic device for subjects with subacute post-stroke UL disability. Furthermore, this is the first study where stroke patients from both treatments completed a TR program at home with total autonomy. This novel approach enabled patients to perform telerehabilitation exercises independently, while still receiving the necessary guidance and oversight from a therapist.

The primary objective is to determine whether this method is safe and if it can effectively reduce perceived disability levels in terms of social participation, activities of daily living, symptom frequency, and the overall impact of disability. Therefore, our main hypothesis is that the proposed TR programs would be safe and effective for motor and cognitive recovery post-stroke. Additionally, we explored the hypothesis that the technological approach (robotic TR) would be at least comparable or superior in improving recovery from subacute post-stroke UL disability compared to non-robotic TR models, given that recent research has demonstrated the potential of robotic interventions in stroke rehabilitation [7,11,28,29,30,31].

The results of this study could have a significant impact on future healthcare policy and practice by demonstrating that subjects with subacute post-stroke UL disability outcomes can receive continuity of care in their home environment through advanced rehabilitation techniques.

## 2. Materials and Methods

### 2.1. Study Design and Participants

We conducted a single-blinded, randomized, two-treatment arm-controlled clinical trial to compare a robotic TR program (referred here as the Robotic Group) with a non-robotic TR program (referred here as the Non-Robotic Group) in subjects with subacute post-stroke UL disability. This study is a part of a multicenter randomized clinical trial involving 90 patients with chronic neurological disease in two rehabilitation centers of the Fondazione Don Carlo Gnocchi, in Italy, from August 2023 to June 2024. The protocol of the trial was previously published [21].

This study is conducted in accordance with the Declaration of Helsinki, the principles of Good Clinical Practice, in accordance with local legislation in participating countries, and approved by the Ethics Committee “Comitato Etico Lazio 1” in May 2023 (Prot. N. 49/CE Lazio 1). The study is registered with ClinicalTrials.gov (NCT06009770).

We included in the analyses subjects (a) affected by an ischemic or hemorrhagic stroke (b) with an upper-limb motor impairment grade > 2 on the Medical Research Council scale (MRC), (c) with age ranging from 25 to 85 years and (d) with cognitive level on a screening test for cognitive impairment (Montreal Cognitive Assessment Test—MoCA Test > 17.54) [32]. We excluded from the study chronic patients (i.e., time since the acute event longer than 6 months), who were unable to sign the informed consent or who had comorbidities that may prevent patients from completing a safe home-based rehabilitation program or lead to clinical instability.

In our facility of the Fondazione Don Carlo Gnocchi in Rome, a total of 119 post-stroke patients with an UL disability were contacted to assess eligibility for the study. Of these, 30 patients met the inclusion criteria, signed the informed consent, and were enrolled in the study. Specifically, 14 patients were randomly assigned to the Non-Robotic Group (NRG), and 16 to the Robotic Group (RG). Figure 1 depicts a diagram of the patients’ progression through the trial (CONSORT diagram).

### 2.2. Intervention

The TR protocol, whether NRG or RG, was performed 4 days a week for 45 min, with one session in synchronous mode (remote connection with the therapist using a telemedicine platform) and the other three in asynchronous mode (the patient performed the exercises provided by the therapist independently, without any remote supervision).

In the context of TR [33], during the synchronous session, the therapist assessed the patient’s progress, verified the correct performance of the exercises for the NRG or exergames for the RG using the end-effector device, and meticulously planned the exercise program for the upcoming week. This served as a critical touchpoint to ensure that the patient remained on track with their rehabilitation goals and to address any difficulties or errors in exercise performance.

While in asynchronous mode, patients performed their exercises autonomously—without real-time supervision from the remotely connected therapist—regardless of their assigned group. These exercises were previously assigned during synchronous sessions with the therapist. More specifically, subjects in the RG completed their three weekly asynchronous sessions using the robot’s end-effector, which recorded their activity in the system. This allowed the therapist to remotely monitor their progress. In contrast, participants in the NRG recorded any difficulties or pain experienced during certain movements in a notebook provided along with the exercise booklet during the three asynchronous sessions. They also noted the days and times they performed the sessions. This information was then reported to the therapist during the teleconference session in the synchronous mode. This measure was in place to ensure compliance with the unsupervised exercises.

A total of 20 sessions were performed in both groups.

Participants of the NRG received digitally supported instructions on the rehabilitation exercises to complete. This exercise booklet included a wide range of activities, such as passive or active-assisted mobilization of the shoulder, elbow, wrist, and hand joints in reaching, grasping, and lifting exercises using common objects, simulating activities of daily living. The RG performed rehabilitation therapy at home using the CE Class II medical device ICONE (Heaxel Srl, Italy). Through planar reaching movements, this end-effector facilitates the recovery of UL disability; in details, the exercises focus on the shoulder and elbow joints. During all exergames, the robot measures (a) the position of its end-effector, controlled by the patient’s hand, and (b) the force exerted by the patient. Feedback is provided through both visual and auditory cues. The feasibility of this approach has already been evaluated in a 2020 study conducted at our rehabilitation center in Rome (submitted).

Figure 2 illustrates the two TR models performed by a patient randomized in the NRG and one in the RG, during a synchronous mode.

For both approaches, the exercises, chosen by the physiotherapists, were adapted on the clinical characteristics of each patient, in terms of UL motor deficit (assessed by the Medical Research Council, MRC) and pain (assessed by the clinical Numeric Rating Scale, NRS) evaluated at baseline. Thus, for both the booklet and the robotic device, three levels of difficulty of the exercises were provided (*easy*, *medium*, *difficult*). The level *easy* was linked to the affected UL’s severe functional impairment, while the level *difficult* was linked to the affected UL’s mild functional impairment.

Specifically, for the NRG’s exercises booklet, as the patient’s clinical condition and the affected UL’s functionality improved, the assistance of the unaffected UL to the affected UL in performing motor tasks decreased while at the difficult level, the task complexity increased.

The same approach is used for the RG, using the end-effector, whose specific protocol based on the level of difficulty is shown in Table 1. This robotic device can function in three different modes: assistive (the motors assist the patient in performing the movement), resistive (the motors act in opposition to the action), and adaptive (the motors automatically adjust the level of support based on the patient’s movements). The assistive and resistive modes can be controlled by the therapist, who sets a parameter ranging from 0 (no assistance/resistance) to 10.

In order to ensure a structured and customized telerehabilitation protocol, a team of experienced robotic physiotherapists studied the best combination of adjustable parameters of the end-effector exergames, adapting them to the progressive evolution of the daily clinical condition of the individual patient recruited in the study, using the tailored method employed in our previous study [16].

Thus, by applying the principles of the four clinical domains (Range of Motion, strength, pain, and spasticity), the specific TR protocol is shown in Table 1.

The exergames’ parameters were selected to manage pain, strength, ROM, and spasticity in the affected UL. For pain, higher levels of assistance and few repetitions were emphasized; for strength, assistance modes were adjusted to avoid exacerbating pain or tone; for ROM, exercises were designed within the maximum tolerated joint excursion; and for spasticity, passive exercises with reduced ROM were used to avoid pain, considering spasticity’s length- and speed-dependence.

### 2.3. Clinical Evaluation

A blinded evaluator evaluated patients at enrollment (Time 0—T0) and at the end of treatment (Time 1—T1).

The primary outcome measure of the study was the change in perceived disability as measured by the World Health Organization Disability Assessment Schedule 2.0 (WHODAS 2.0) in each time point of the study (T0 and T1). The WHODAS 2.0 [34] questionnaire assesses functioning and disability in six domains (cognition, mobility, self-care, getting by, life activities, and participation in community activities) using the International Classification of Functioning, Disability, and Health (ICF) [35], and participants are asked to rate their difficulty in performing various daily activities in relation to any health condition.

Secondary outcomes were changes in motor, pain, and cognitive measures and changes in mood sphere clinical evaluations, assessed at each evaluation time point (T0 and T1).

UL motor functions was specifically evaluated using:

the Fugl-Meyer Assessment for Upper Extremity (FMA-UE), a performance-based impairment index that is specific to stroke. The UL maximum motor score is 66, with each item being scored on a 3-point ordinal scale [36];

the Action Research Arm Test (ARAT) is a 19-item test that assesses the functionality, dexterity, and coordination of the UL. Higher scores indicate better performance. Items are scored on a 4-point ordinal scale, with scores ranging from 0 to 57 [37];

the Box and Block Test (BBT) assesses gross manual dexterity by asking the patient to transfer as many blocks as possible from one compartment to another, one at a time, in 60 s. A higher score indicates greater manual dexterity [38].

UL pain measure was specifically evaluated using:

the Numeric Rating Scale (NRS) scale is an 11-point numerical scale with segments ranging from 0 to 10. Patients are asked to choose an appropriate number to express the UL level of pain they are experiencing [39].

Cognitive functions were specifically evaluated using:

the Montreal Cognitive Assessment (MoCA) is a battery of screening questions that includes subtests to assess abstraction, set shifting, and cognitive flexibility (MoCA total score range: 0–30). Higher scores indicate better overall cognitive performance [40];

the Trail Making Test (TMT) is a neuropsychological test consisting of a dual-task (TMT-B) and a visual scanning (TMT-A) component. The time required to complete each section of the test determines the TMT score. In addition, the difference in time between TMT-B and TMT-A is measured to assess executive functioning. Poor performance is indicated by long execution times [41];

the Symbol Digit Modalities Test (SDMT) is a commonly used test of psychomotor speed. It requires participants to replace a series of digits with geometric symbols in relation to a given digit-symbol key in a limited time (90 s). Higher scores indicate better performance [42].

Mood sphere clinical evaluations were:

the Beck Depression Inventory (BDI) is a seven-item test with a 0–3-point scoring system. The scores for each item are added together to give a total score (range 0–21). Greater deflection of the mood tone is indicated by higher scores [43];

the State Trait Anxiety Inventory—Y1 (STAI-Y1) is a commonly used measure of trait and state anxiety (20 items each). All items are scored on a 4-point scale (from ‘almost never’ to ‘almost always’). Higher scores indicate greater anxiety [44].

### 2.4. Statistical Analysis

The Mann–Whitney U test (for numerical variables) and the Chi-squared test (for categorical variables) were used to compare baseline characteristics of the two groups.

To compare data of all outcome measures obtained at T0 and T1 between groups, a mixed Analysis of Variance (ANOVA) statistical test was conducted, with the randomization group variable (two levels: Non-Robotic vs. Robotic) as the between-group factor, and time variable (two levels: T0 vs. T1) as the within-group factor. Assumption of equality of variances was confirmed by means of the Levene’s test.

Statistical analyses were performed using the SPSS Statistics software (version 28, IBM Corp., Armonk, NY, USA). A p-value lower than 0.05 was considered as significant in all the analyses.

Statistical analyses were performed by an investigator with no clinical role in the study and blinded to the randomization groups.

## 3. Results

One hundred nineteen subjects with subacute post-stroke UL disability were assessed for eligibility. Of them, 72 were excluded because of the inclusion criteria, four declined to participate, and 13 because of other reasons. Therefore, 30 subjects (NRG: n = 14; RG: n = 16) were included in the analysis.

No dropouts or missing data were registered during the study.

The demographic and clinical characteristics at baseline of the participants, together with related statistical comparison at T0, are shown in Table 2. Statistical analysis revealed that the two randomized groups shared similar characteristics and were therefore comparable.

During the study, no adverse events were observed.

With respect to the primary outcome of disability, namely the WHODAS 2.0, the results of the mixed ANOVA test showed that the interaction factor timeXgroup was not significant, while the main effect time was statistically significant (*p* = 0.045, estimated effect size η^2^ = 0.013). In particular, the WHODAS 2.0 showed a mean decrease from baseline of 6.09 ± 2.62% in the NRG, compared to a mean decrease of 0.76 ± 2.21% in the RG.

With respect to the secondary outcomes, the results of the mixed ANOVA test showed:
a statistically significant interaction factor timeXgroup on the FMA-UE motor function (*p* = 0.006, η^2^ = 0.006), the BBT (*p* = 0.036, η^2^ = 0.148), and the TMT-A (*p* = 0.018, η^2^ = 0.011);a statistically significant main effect time on the FMA-UE motor function (*p* < 0.001, η^2^ = 0.033), the FMA-UE sensation (*p* < 0.001, η^2^ = 0.112), the ARAT (*p* < 0.001, η^2^ = 0.014), the BBT (*p* = 0.004, η^2^ = 0.263), the MoCA (*p* = 0.005, η^2^ = 0.027), the TMT-A (*p* < 0.001, η^2^ = 0.024), the TMT-B (*p* = 0.001, η^2^ = 0.032), and the BDI (*p* = 0.018, η^2^ = 0.039);no statistical relevance on NRS, SDMT, and STAY-Y1.

Regarding the statistically significant improvements, in the NRG, the FMA-UE motor function showed a mean increase of 10.79 ± 2.17 points from baseline, compared to an increase of 4.44 ± 0.79 points in the RG. The FMA-UE sensation showed a mean increase of 2.29 ± 0.89 points from baseline in the NRG, compared to an increase of 2.0 ± 0.70 points in the RG. The ARAT showed a mean improvement of 7.71 ± 2.05 points in the NRG, compared to an improvement of 3.25 ± 1.68 points in the RG. The BBT (related to affected size) showed a mean improvement of 5.57 ± 1.72 points in the NRG, compared to an improvement of 1.0 ± 1.22 points in the RG. Cognitive function, as measured by the TMT-A, improved with a decrease from baseline of 5.86 ± 4.39 s in the NRG, compared to a decrease from baseline of 28.69 ± 7.57 s in the RG. The TMT-B showed instead a decrease from baseline of 31.43 ± 9.85 s in the NRG, compared to a decrease from baseline of 26.06 ± 12.28 s in the RG. The MoCA showed a mean change from baseline of 2.14 ± 0.62 in the NRG, and a mean change from baseline of 0.75 ± 0.70 in the RG. The BDI showed a mean decrease from baseline of 2.64 ± 1.70 points in the NRG, and a mean decrease from baseline of 4.94 ± 2.39 in the RG.

Figure 3 shows results of the different clinical evaluations (primary and secondary outcomes) performed at T0 and T1 for the two intervention groups, together with the statistical significance related to the main effect time and the interaction timeXgroup factor.

## 4. Discussion

The aim of this study was to verify the safety and the effectiveness of a TR program in subjects with subacute post-stroke UL disability, completing home-based rehabilitation activities alone while obtaining the required advice and supervision only once a week from a physiotherapist.

The TR intervention, delivered using either a non-robotic approach or a robotic device, has been confirmed to be a safe approach for stroke survivors [45,46] with no adverse events reported during or after treatment. All 30 post-stroke participants completed the 20-session TR program without any problems experienced with the robotic device or the telemedicine platform. Moreover, recruited subjects reported a decrease in anxiety and depression and no increase in pain.

Our results are in line with previous literature showing the non-inferiority of the TR intervention for stroke survivors [46,47,48]. Specifically, patients in our study show not only a significant statistical improvement in motor performance, already demonstrated by Tchero et al., Su et al., and other trials [46,47], but also in sensory and cognitive domains, as well as in the disability and mood spheres in both groups. In accordance, a recent systematic review demonstrated that physiotherapy interventions using TR were at least as successful as traditional rehabilitation interventions and should be considered a feasible and effective solution [49].

An interesting finding was that the treatment provided in the NRG showed a greater effect, compared to the RG, as measured by the FMA-UE motor function, BBT, and TMT-A. This could be because the TR program for the NRG was developed by a team of highly experienced physical therapists specializing in upper extremity rehabilitation, and it included more specific motor tasks and a wider range of activities. In contrast, the rehabilitation tasks provided by the robotic device focused primarily on reaching exergames in the horizontal plane. In accordance, Morone and Pichiorri [50] suggest that a well-structured rehabilitation program, tailored to the subjects’ characteristics, and combined with expert supervision using the TR platform, can have a significant impact, sometimes even exceeding the benefits of using robotic technologies alone.

In this study, while several outcomes showed statistically significant improvements in both the Robotic Group (RG) and the Non-Robotic Group (NRG), certain measures did not reach statistical significance. For example, the improvement in pain levels (as measured by the Numeric Rating Scale) and anxiety (as measured by the STAI-Y1) was not statistically significant in the two groups. The lack of significant differences in these domains may be attributed to several factors. First, the relatively small sample size may have limited the power of the study to detect subtle differences between the groups and changes within the groups, particularly for secondary outcomes such as pain and anxiety. Second, the duration of the intervention (20 sessions over 5 weeks) may have been sufficient to improve motor and cognitive function but too short to significantly impact mood and pain outcomes, which may require longer-term interventions.

In regard to the symptom of pain, the aforementioned systematic review indicates that only one study [51] has investigated this phenomenon. Given the importance of pain as a symptom in the context of rehabilitation, several of our trials [7,16], have included pain as one of four primary patient characteristics on which the treatment protocol was tailored. Pain is a common barrier to rehabilitation adherence, and in stroke recovery, excessive pain can delay or prevent functional improvements [52]. It is important to note that no increase in pain was observed after the intervention, as indicated by the NRS. In the context of home-based, unsupervised telerehabilitation, it is of paramount importance to ascertain that patients do not experience an exacerbation of pain, as this serves as a crucial indicator of the safety and feasibility of the intervention. Conversely, the low significance observed at T1 may be attributed to the relatively limited sample size or the inclusion of subjects who exhibited no discernible pain at the baseline assessment [53].

Both methods were delivered in a home setting, following a TR program under the remote supervision of an experienced therapist, using a telemedicine platform for the videoconference once a week while the other three times a week without any supervision. The combination of professional oversight with patient autonomy not only empowered patients but also provided them with the flexibility to manage their rehabilitation in a way that best suited their personal schedules and needs. This innovative approach could lead to improved adherence, better functional outcomes, and an overall enhancement in the quality of life for stroke survivors [54].

In literature, different studies have reported a TR approach for patients with subacute post-stroke disabilities: however, all of these are based on a full supervised exercise program using different technology tools such as videoconference systems, a cloud-telehealth system, telemedicine platforms, and others to monitor the therapy session [26,46,47,54,55,56].

Only a few studies described a similar TR approach but performed by subjects with low back pain of other musculoskeletal disorders [51,57,58]. The structure of our study allowed patients to integrate rehabilitation into their daily routines more seamlessly, fostering a sense of ownership and responsibility over their recovery process [33].

In this direction, the experience gained from our study was instrumental in defining the TR model of a multicenter study protocol using technology tools, with the aim of evaluating the effectiveness of the UL treatment program, as performed in our study from the NRG, in an enhanced home environment using digital technologies in a large cohort of patients with subacute post-stroke UL disability (STROKEFIT4HOME pragmatic trial). Such an approach would combine technology with professional expertise to develop UL-specific TR protocols. Recording feedback from such sensor-equipped objects [56], ref. [59] could enable continuous monitoring of treatment and would allow the physiotherapists to tailor the treatment to the patient’s needs, even remotely. In addition, the use of these technologies would facilitate the collection of rich data that could be analyzed to continuously improve rehabilitation protocols. This data, combined with traditional clinical assessments, could provide a comprehensive view of the patient’s progress, enabling dynamic and personalized treatment adjustments.

This study enabled us to explore an innovative remote therapy protocol for UL rehabilitation in home settings. The study underscores the importance of accessible healthcare solutions, especially for individuals with limited mobility. By utilizing remote monitoring tools, we ensured continuous support and progress tracking.

The positive outcomes suggest that a TR program can complement traditional rehabilitation methods. Further research is needed to refine these protocols and assess long-term efficacy. Overall, this approach has the potential to completely transform the way we provide therapeutic services by making them more flexible and inclusive.

### Limitations

The principal limitation of this study is the relatively small sample size. Furthermore, the absence of long-term follow-up precludes an evaluation of the sustainability of the observed improvements in both groups. Future studies should aim to address these limitations by enrolling larger sample sizes and incorporating longer follow-up periods.

## 5. Conclusions

Our study shows that TR can enhance rehabilitation by improving motor and cognitive skills, patient autonomy, and protocol adherence in subjects with subacute UL disability. Both models of TR, non-robotic and robotic, offer safety and significant benefits, with the NRG approach demonstrating significantly greater effectiveness due to its specificity and variability. The future of home rehabilitation may lie in the combination of advanced technologies with professional expertise, using sensor-equipped objects and telerehabilitation platforms.

## Figures and Tables

**Figure 1 brainsci-14-00941-f001:**
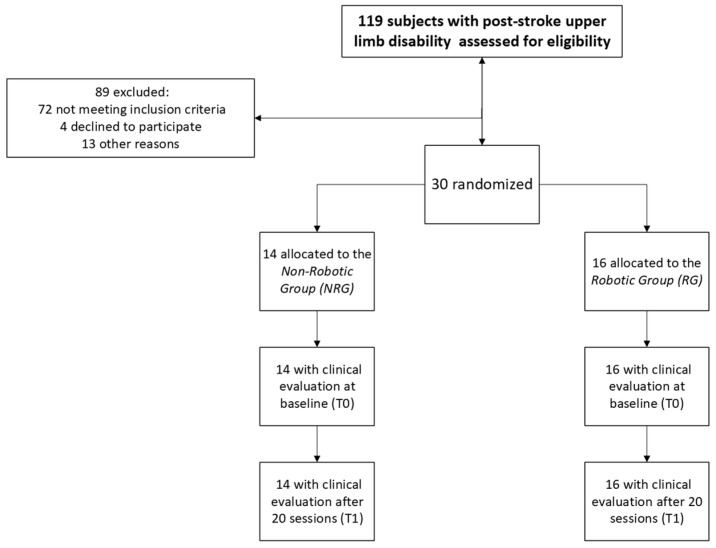
CONSORT diagram showing the different phases of the study.

**Figure 2 brainsci-14-00941-f002:**
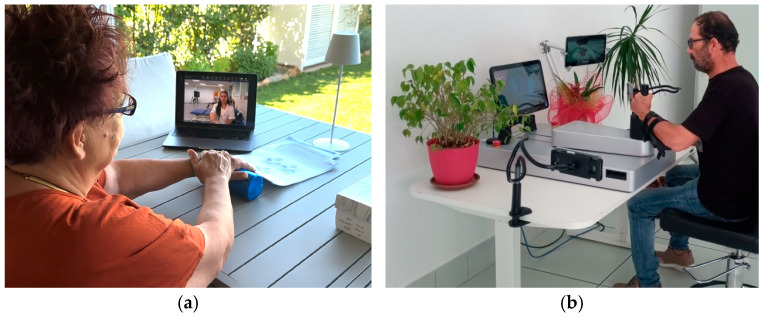
The two telerehabilitation models. (**a**) A patient randomized to the Non-Robotic Group performs upper limb exercises remotely connected to the therapist; (**b**) A patient randomized to the Robotic Group performs upper limb exercises using the end-effector remotely connected to the therapist.

**Figure 3 brainsci-14-00941-f003:**
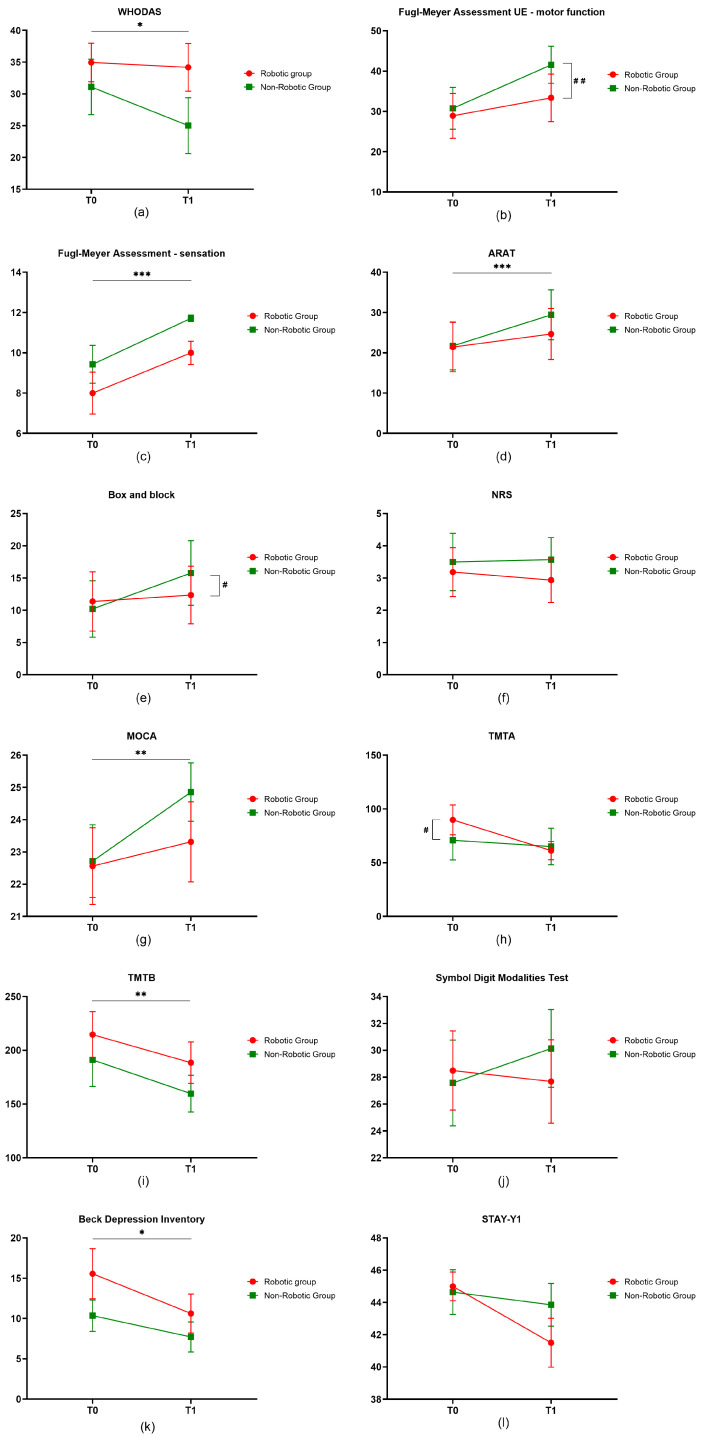
Study outcome at T0 and T1. The graphs for the different scales evaluated are shown in the figures above: (**a**) WHODAS. A statistically significant decrease on the scale indicates a decrease in disability with the treatment (*: *p* < 0.05); (**b**) Fugl-Meyer Assessment UE—motor function. The statistically significant increase on the scale indicates an improvement in motor performance. A difference in behavior between the two groups was also observed (##: *p* < 0.01); (**c**) Fugl-Meyer Assessment UE—sensation. The statistically significant increase on the scale indicates an improvement in sensitivity (***: *p* < 0.001); (**d**) Action Research Arm Test (ARAT). The statistically significant increase on the scale indicates an improvement in motor performance (***: *p* < 0.001); (**e**) Box and Block Test. The statistically significant increase on the scale indicates an improvement in dexterity. Different behavior was also observed between the two groups (#: *p* < 0.05); (**f**) Numerical Rating Scale (NRS). A decrease on the scale indicates a decrease in pain; (**g**) Montreal Cognitive Assessment (MoCA). The statistically significant increase on the scale indicates improvement in cognitive function (**: *p* < 0.01); (**h**) Trail Making Test-A (TMTA). The statistically significant decrease in the test indicates improvement in visual and spatial search tasks. Differences in behavior between the two groups were also observed (#: *p* < 0.05); (**i**) Trail Making Test-B (TMTB). The statistically significant decrease in the test indicates improvement in visual and spatial search tasks (**: *p* < 0.01); (**j**) Symbol Digit Modalities Test (SDMT). An increase on the scale indicates an improvement in attention and working memory; (**k**) Beck Depression Inventory (BDI). A statistically significant decrease in the test indicates a reduction in depression (*: *p* < 0.05); (**l**) STAY-Y1. A decrease on the scale indicates a decrease in anxiety.

**Table 1 brainsci-14-00941-t001:** Specific telerehabilitation protocols using the end-effector robot in the Robotic Group based on the patient’s clinical characteristics.

Deficit/Pain	Parameters	Exergame	Exergame’s Repetitions	Repetitions’ Series
Severe impairment	Exergames with an assistance level between 6 and 8	All from the library	16	2
Moderate impairment	Exergames with an assistance level between 3 and 5	All from the library	32	2
Mild impairment	Exergames with an assistance level between 0 and 2 or exercises in adaptive mode	All from the library	32	2
Pain ≤ 4	Exergames, regardless of UL impairment, with assistance greater than 6	All from the library	16	2

**Table 2 brainsci-14-00941-t002:** Demographic characteristics of the study’s subjects with subacute post-stroke upper limb disability divided into the two treatment groups. Numbers shown are means (± standard deviations) or cases (percent). The *p*-value refers to the statistical performed via the Mann–Whitney U-test for numerical variables or the Chi-squared test for categorical variables. A *p*-value greater than 0.05 indicates no statistically significant difference between the two groups.

		Non-Robotic Group (n = 14)	Robotic Group (n = 16)	*p*-Value
Age (years)		65.7 (±13.3)	65.7 (±11.3)	0.995
Sex	Men	6 (42.9%)	11 (68.8%)	0.153
	Women	8 (57.1%)	5 (31.3%)	
Latency (days)		122.7 (±73.9)	126.8 (85.3)	0.890
Schooling (years)		11.1 (±3.2)	9.5 (±3.2)	0.164
Occupation	Employed	1 (7.1%)	1 (6.3%)	0.922
	Unemployed/retired	13 (92.2%)	15 (93.8%)	
Primary outcome	World Health Organization Disability Assessment Schedule 2.0	31.1 (16.3)	34.9 (12.1)	0.678
Secondary outcomes	Fugl-Meyer Assessment for Upper Extremity-motor function	30.8 (19.5)	28.9 (22.3)	0.677
	Fugl-Meyer Assessment for Upper Extremity-sensation	9.4 (3.5)	8.0 (4.1)	0.495
	Action Research Arm Test	21.7 (22.2)	21.4 (24.5)	1.000
	Box and Block Test-Right	31.2 (21.8)	38.9 (21.0)	0.560
	Box and Block Test-Left	23.4 (22.9)	17.8 (22.7)	0.430
	Numeric Rating Scale	3.5 (3.3)	3.2 (3.0)	0.780
	Montreal Cognitive Assessment	22.7 (4.2)	22.6 (4.8)	0.950
	Trail Making Test-A	70.9 (68.8)	89.9 (55.4)	0.129
	Trail Making Test-B	191.1 (92.7)	214.6 (85.8)	0.349
	Symbol Digit Modalities Test	27.6 (11.9)	28.5 (11.8)	1.000
	Beck Depression Inventory	10.4 (7.3)	15.6 (12.4)	0.211
	State Trait Anxiety Inventory—Y1	44.6 (5.2)	45.0 (3.6)	0.723

## Data Availability

The data that support the findings of this study are available from the corresponding author, upon reasonable request.

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
