# Peer review of "Effectiveness of Two Models of Telerehabilitation in Improving Recovery from Subacute Upper Limb Disability after Stroke: Robotic vs. Non-Robotic"

_brainsci, 2024, doi:10.3390/brainsci14090941_

Round 1

Reviewer 1 Report

Comments and Suggestions for Authors

Dear Authors,

Thank you for submitting your manuscript on the effectiveness of telerehabilitation for upper limb recovery after stroke. The study tackles a highly relevant issue in stroke rehabilitation and presents valuable insights into the use of both robotic and non-robotic models. After reviewing your manuscript, I have the following suggestions:

Lines 9–26: The abstract lacks specificity regarding the statistical results, which would strengthen the conclusion. Adding more details on the exact improvements seen in each group would provide clarity.

Lines 30–80: The introduction is missing a clear statement of the hypothesis or research question, which would frame the study more clearly. It could also benefit from a more in-depth review of recent studies on robotic rehabilitation and non-robotic interventions in stroke recovery to support the study rationale.

Lines 82–120: The sample size justification is missing. There is no power analysis to determine whether the sample size of 30 is sufficient to detect meaningful differences between groups.

Lines 108–118: More details are needed regarding the asynchronous mode in the non-robotic group. For example, what measures were in place to ensure compliance with the unsupervised exercises?

 Lines 213–223: The statistical analysis could benefit from a more comprehensive explanation of how missing data or drop-outs were handled, if applicable.

Lines 224–245: While the key results are reported, there is a lack of detailed reporting on individual group differences. For example, the specific magnitude of changes within each group, particularly in the robotic group, could be better elaborated.

Lines 229–233: There is no mention of effect sizes, which are important for understanding the practical significance of the findings beyond statistical significance Lines 246–281: More detailed legends and explanations for the figures would enhance clarity. Additionally, some figures could benefit from more descriptive titles to aid interpretation.

Lines 282–353: The discussion lacks depth regarding the limitations of the study. For instance, the small sample size, potential biases in self-reported measures, or lack of long-term follow-up should be acknowledged more explicitly. The discussion could also explore the implications of the non-significant findings more thoroughly.

Thank you

Author Response

Comments 1: Lines 9–26: The abstract lacks specificity regarding the statistical results, which would strengthen the conclusion. Adding more details on the exact improvements seen in each group would provide clarity.

Response 1: We thank the reviewer for pointing this out. We have modified the abstract in order to clarify which clinical scales’ data presented significant differences in behavior between the groups. Furthermore, we have detailed the exact improvements of the groups in the primary outcome measure. The abstract now reads:

Background/Objectives: Finding innovative digital solutions is fundamental to ensure prompt and continuous care for patients with chronic neurological disorders, whose demand for rehabilitation also in home-based settings is steadily increasing. The aim is to verify the safety and the effectiveness of two telerehabilitation (TR) models in improving recovery from subacute upper limb disability after stroke, with and without a robotic device. Methods: One hundred nineteen subjects with subacute post-stroke upper limb disability were assessed for eligibility. Of them, thirty patients were enrolled in the study and randomly assigned to either the Robotic Group (RG), undergoing a 20-session TR program, using a robotic device, or the Non-Robotic Group (NRG) undergoing a 20-session TR program without robotics. Clinical evaluations were measured at baseline (T0) and post-intervention (T1, 5 weeks after baseline) and included assessments of quality of life, motor skills, and clinical/functional status. The primary outcome measure was the World Health Organization Disability Assessment Schedule 2.0, evaluating the change in perceived disability. Results: Statistical analysis shows that patients of both groups improved significantly over time in all domains analyzed (mean decrease from baseline in the WHODAS 2.0 of 6.09 ± 2.62% for the NRG, and of 0.76 ± 2.21% for the RG), with a greater improvement of patients in the NRG in motor (Fugl-Meyer Assessment Upper Extremity - motor function, Box and Block Test) and cognitive skills (Trail Making Test-A). Conclusions: This study highlights the potential of TR programs to transform stroke rehabilitation by enhancing accessibility and patient-centered care, promoting autonomy, improving adherence, and leading to better outcomes and quality of life for stroke survivors.”

Comments 2: Lines 30–80: The introduction is missing a clear statement of the hypothesis or research question, which would frame the study more clearly. It could also benefit from a more in-depth review of recent studies on robotic rehabilitation and non-robotic interventions in stroke recovery to support the study rationale.

Response 2: We agree with the Reviewer’s point. We have, accordingly, modified the text of the Introduction (lines 92-98, p.2) to clearly state our study’s hypotheses:

“[…] Therefore, our main hypothesis is that the proposed TR programs would be safe and effective for motor and cognitive recovery post-stroke. Additionally, we explored the hypothesis that the technological approach (robotic TR) would be at least comparable or superior in improving recovery from subacute post-stroke UL disability compared to non-robotic TR models, given that recent research has demonstrated the potential of robotic interventions in stroke rehabilitation”.

Furthermore, we have inserted a more in-depth review of recent studies on robotic rehabilitation and non-robotic interventions in stroke recovery to support the study rationale. The text now reads (line 53-70, p.2):

“[…] Robotic rehabilitation for individuals with subacute upper limb (UL) disability following a stroke is increasingly endorsed as an adjunct to conventional therapy in select stroke guidelines. These robotic systems are viewed as a promising addition, offering repetitive and precise movements that may enhance recovery. However, despite growing interest, scientific evidence comparing the efficacy of robotic rehabilitation to traditional approaches remains inconclusive. Some studies show improvements in motor function and faster recovery with robotics, while others report minimal differences. The variability in patient responses and rehabilitation protocols complicates drawing definitive conclusions. Thus, while robotic rehabilitation represents an exciting advancement, its full potential compared to conventional therapy is still under investigation. Robotic systems have proven effective in supporting the recovery of motor and cognitive function in stroke patients. Large randomized clinical trials have demonstrated their equivalence to traditional rehabilitation when a similar treatment dose is provided. Additionally, these technologies enable the intensification of treatment and the development of personalized treatment plans based on objective and measurable out-comes, especially in improving the recovery from subacute UL disability after stroke”.

Comments 3: Lines 82–120: The sample size justification is missing. There is no power analysis to determine whether the sample size of 30 is sufficient to detect meaningful differences between groups.

Response 3: We thank the Reviewer for the observation. As we specify in the manuscript this study is a part of a multicenter randomized clinical trial involving ninety patients with chronic neurological disease in two rehabilitation centers of the Fondazione Don Carlo Gnocchi, in Italy, from August 2023 to June 2024 (line 108-111, p.3, “Materials and methods” section). The protocol of the trial was previously published. In particular, the power-based estimation of the sample size was computed for the cited multicenter study. According to an a-priori sample size calculation (using the G*Power software), 90 subjects (45 subjects per arm), comprehensive of up to 10% drop-off, is sufficient to detect a medium to large standardized effect size (0.695) when considering the independent comparison between two groups (two sided unpaired t-test). The effect size was obtained from our previous unpublished study that compared two groups of treatments in WHODAS 2.0 (Federici et al., 20217, doi: 10.1080/09638288.2016.1223177) total score and was chosen as a benchmark as the studies are similar in terms of methods and materials. Power calculation was conducted considering a type-I error rate of 5% (α = 0.05) with a statistical power of 0.80. The study is registered with ClinicalTrials.gov (NCT06009770), as reported in the text, line 115, p.3, “Materials and methods” section. Should the reviewer determine that this is a beneficial addition, we can incorporate it into the manuscript text.

Comments 4: Lines 108–118: More details are needed regarding the asynchronous mode in the non-robotic group. For example, what measures were in place to ensure compliance with the unsupervised exercises?

Response 4: We apologize for the lack of clarity in this aspect. We have now modified the manuscript (line 142-153, p.3/4, “Intervention” section) as follows:

“[…] In asynchronous mode, patients performed their exercises autonomously—without real-time supervision from the remotely connected therapist—regardless of their assigned group. These exercises were previously assigned during synchronous sessions with the therapist. More specifically, subjects in the RG completed their three weekly asynchronous sessions using the robot's end-effector, which recorded their activity in the system. This allowed the therapist to remotely monitor their progress. In contrast, participants in the NRG recorded any difficulties or pain experienced during certain movements in a note-book provided along with the exercises booklet during the three asynchronous sessions. They also noted the days and times they performed the sessions. This information was then reported to the therapist during the teleconference session in the synchronous mode. This measure was in place to ensure compliance with the unsupervised exercises.”

Comments 5: Lines 213–223: The statistical analysis could benefit from a more comprehensive explanation of how missing data or drop-outs were handled, if applicable.

Response 5: We apologize for the lack of clarity in this aspect. We have modified the line 266 (p.6, “Results” section) of the manuscript as follows:

“[…] No drop-outs or missing data were registered during the study.

Comments 6: Lines 224–245: While the key results are reported, there is a lack of detailed reporting on individual group differences. For example, the specific magnitude of changes within each group, particularly in the robotic group, could be better elaborated.

Response 6: We thank the Reviewer for the observation. We have now inserted in the “Results” section the magnitudes of changes from baseline for the evaluated scales. The text now reads:

Lines 272-276, p.6: “With respect to the primary outcome of disability, namely the WHODAS 2.0, the results of the mixed ANOVA test showed that the interaction factor timeXgroup was not significant, while the main effect time was statistically significant (P=0.045, estimated effect size η2=0.013). In particular, the WHODAS 2.0 showed a mean decrease from baseline of 6.09 ± 2.62 % in the NRG, compared to a mean decrease of 0.76 ± 2.21% in the RG.

Lines 288-302, p.6: “Regarding the statistically significant improvements, in the NRG, the FMA-UE motor function showed a mean increase of 10.79 ± 2.17 points from baseline, compared to an increase of 4.44 ± 0.79 points in the RG. The FMA-UE sensation showed a mean increase of 2.29 ± 0.89 points from baseline in the NRG, compared to an increase of 2.0 ± 0.70 points in the RG. The ARAT showed a mean improvement of 7.71 ± 2.05 points in the NRG, compared to an improvement of 3.25 ± 1.68 points in the RG. The BBT (related to affected size) showed a mean improvement of 5.57 ± 1.72 points in the NRG, compared to an improvement of 1.0 ± 1.22 points in the RG. Cognitive function, as measured by the TMT-A, improved with a decrease from baseline of 5.86 ± 4.39 seconds in the NRG, compared to a decrease from baseline of 28.69 ± 7.57 seconds in the RG. The TMT-B showed instead a decrease from baseline of 31.43 ± 9.85 seconds in the NRG, compared to a decrease from baseline of 26.06 ± 12.28 seconds in the RG. The MoCA showed a mean change from baseline of 2.14 ± 0.62 in the NRG, and a mean change from baseline of 0.75 ± 0.70 in the RG. The BDI showed a mean decrease from baseline of 2.64 ± 1.70 points in the NRG, and a mean decrease from baseline of 4.94 ± 2.39 in the RG.

Comments 7: Lines 229–233: There is no mention of effect sizes, which are important for understanding the practical significance of the findings beyond statistical significance.

Response 7: We apologize for the lack of details. We have now inserted in the “Results” section the estimated effect sizes (η2). The text now reads:

Lines 272-287, p.6: “With respect to the primary outcome of disability, namely the WHODAS 2.0, the results of the mixed ANOVA test showed that the interaction factor timeXgroup was not significant, while the main effect time was statistically significant (P=0.045, estimated effect size η2=0.013).

[…]

With respect to the secondary outcomes, the results of the mixed ANOVA test showed:

·         a statistically significant interaction factor timeXgroup on the FMA-UE motor function (P=0.006, η2=0.006), the BBT (P=0.036, η2=0.148) and the TMT-A (P=0.018, η2=0.011);

·         a statistically significant main effect time on the FMA-UE motor function (P<0.001, η2=0.033), the FMA-UE sensation (P<0.001, η2=0.112), the ARAT (P<0.001, η2=0.014), the BBT (P=0.004), the MoCA (P=0.005, η2=0.027), the TMT-A (P<0.001, η2=0.024), the TMT-B (P=0.001, η2=0.032) and the BDI (P=0.018, η2=0.039);

·         no statistical relevance on NRS, SDMT and STAY-Y1.

Comments 8: Lines 246–281: More detailed legends and explanations for the figures would enhance clarity. Additionally, some figures could benefit from more descriptive titles to aid interpretation.

Response 8: We apologize for the lack of details. We have now inserted:

·         Line 310, p.7, Figure 2: “The two telerehabilitation models”;

·         Line 314, p.7, Table 1: “Specific telerehabilitation protocol using the end-effector robot in the Robotic Group based on patient’s clinical characteristics”;

·         Line 317, p.8, Table 2: “Demographic characteristics of study’s subjects with subacute post stroke upper limb disability divided into the two treatment groups. Numbers shown are means (± standard deviations) or cases (percent). The p-value refers to the statistical performed via the Mann-Whitney U-test for numerical variables, or the Chi-squared test for categorical variables. A p-value greater than 0.05 indicates no statistically significant difference between the two groups.”

·         Line 323, p.9, Figure 3: “Study outcome at T0 and T1”

Comments 9: Lines 282–353: The discussion lacks depth regarding the limitations of the study. For instance, the small sample size, potential biases in self-reported measures, or lack of long-term follow-up should be acknowledged more explicitly. The discussion could also explore the implications of the non-significant findings more thoroughly.

Response 9: We understand the Reviewer’s concerns. We now have both expanded the “Discussion” section including implications of the non-significant findings (p.10/11) and included a new section 4.1 named “Limitations” (p. 12) in the manuscript. The text now reads:

Lines 372-394: “In this study, while several outcomes showed statistically significant improvements in both the Robotic Group (RG) and Non-Robotic Group (NRG), certain measures did not reach statistical significance. For example, the improvement in pain levels (as measured by the Numeric Rating Scale) and anxiety (as measured by the STAI-Y1) was not statistically significant in the two groups. The lack of significant differences in these domains may be attributed to several factors. First, the relatively small sample size may have limited the power of the study to detect subtle differences between the groups and changes within the groups, particularly for secondary outcomes such as pain and anxiety. Second, the duration of the intervention (20 sessions over five weeks) may have been sufficient to improve motor and cognitive function but too short to significantly impact mood and pain outcomes, which may require longer-term interventions.

In regard to the symptom of pain, the aforementioned systematic review indicates that only one study has investigated this phenomenon. Given the importance of pain as a symptom in the context of rehabilitation, several of our trials have included pain as one of four primary patient characteristics on which the treatment protocol was tailored. Pain is a common barrier to rehabilitation adherence, and in stroke recovery, excessive pain can delay or prevent functional improvements. It is important to note that no increase in pain was observed after the intervention, as indicated by the NRS. In the context of home-based, unsupervised telerehabilitation, it is of paramount importance to ascertain that patients do not experience an exacerbation of pain, as this serves as a crucial indicator of the safety and feasibility of the intervention. Conversely, the low significance observed at T1 may be attributed to the relatively limited sample size or the inclusion of subjects who exhibited no discernible pain at the baseline assessment.

Lines 433-437: “4.1. Limitations

The principal limitation of this study is the relatively small sample size. Furthermore, the absence of long-term follow-up precludes an evaluation of the sustainability of the observed improvements in both groups. Future studies should aim to address these limitations by enrolling larger sample sizes and incorporating longer follow-up periods.

In regard to self-reported measures, the WHODAS scale was selected as the primary outcome of the study, with the objective of measuring the change in perceived disability across six domains (cognition, mobility, self-care, independence in daily activities, life activities, and participation in community activities) in accordance with the International Classification of Functioning, Disability, and Health (ICF). A recent systematic review (Potcovaru et al., 2024, doi:10.3390/jcm13051252) indicates that WHODAS 2.0 is a useful tool for assessing health and disability levels in the general population via surveys and for measuring the effectiveness and productivity of interventions. However, the most accurate results are obtained when the WHODAS 2.0 is used in conjunction with other evaluation methods. Indeed, in our study, we selected more specific clinical assessments as secondary outcomes in order to more accurately reveal the improvement in recovery from subacute upper limb disability following a stroke.

Reviewer 2 Report

Comments and Suggestions for Authors

Interesting paper with useful rehabilitation protocol.

I have only few minor edits to the paper:

1. Pleas revise the title accordingly: 

Effectiveness of two models of telerehabilitation in improving recovery from subacute upper limb disability after stroke: robotic vs. non-robotic 

Key words: please check the MeSH https://www.ncbi.nlm.nih.gov/mesh/ and revise

13 for upper limb recovery in patients /should be: in improving recovery from subacute upper limb disability after stroke

14 hundred nineteen subjects with subacute  stroke were assessed for eligibility/ should be: subacute post stroke  upper limb disability

85 subjects with stroke/ should be: subjects with subacute post stroke  upper limb disability - please change it thruoughout the paper

228 drop out/ drop- outs

figure 1 pazients/ subjects with subacute post stroke  upper limb disability

table 1 and table 2 missing titles

discussion: mising limitations of the study whish is very importan to state

Author Response

Comments 1: Pleas revise the title accordingly:

Effectiveness of two models of telerehabilitation in improving recovery from subacute upper limb disability after stroke: robotic vs. non-robotic

Response 1: Thank you for pointing this out. We agree with this comment. Therefore, we have modified the title (line 2-4, p.1) as follows:

“Effectiveness of two models of telerehabilitation in improving recovery from subacute upper limb disability after stroke: robotic vs. non-robotic”

Comments 2: Key words: please check the MeSH https://www.ncbi.nlm.nih.gov/mesh/ and revise

Response 2: We thank the Reviewer for the observation. We have, accordingly, revised and modified some key words to emphasize this point as follows:

Line 30-31, p.1: ”stroke; rehabilitation; telehealth; home-based; upper extremity; robotics; robot-assisted rehabilitation training; virtual reality; rehabilitation protocol.”

Comments 3: line 13 for upper limb recovery in patients /should be: in improving recovery from subacute upper limb disability after stroke

Response 3: We thank the Reviewer for the observation. We have now modified the line 14-15 (p.1, “Abstract” section) of the manuscript as follows:

”[…] in improving recovery from subacute upper limb disability after stroke”.

Comments 4: line 14 hundred nineteen subjects with subacute  stroke were assessed for eligibility/ should be: subacute post stroke  upper limb disability

Response 4: We thank the Reviewer for the observation. We have now modified the line 15-16 (p.1, “Abstract” section) of the manuscript as follows:

”One hundred nineteen subjects with subacute post-stroke upper limb disability were assessed for eligibility”.

Comments 5: line 85 subjects with stroke/ should be: subjects with subacute post stroke  upper limb disability - please change it throughout the paper

Response 5: We thank the Reviewer for the observation. We have now modified the line 100 (p.3, “Introduction” section) of the manuscript as follows:

”[…] in subjects with subacute post-stroke UL disability”.

Furthermore, we have modified it throughout the manuscript.

Comments 6: 228 drop out/ drop- outs               

Response 6: We thank the Reviewer for the observation. We have now modified the line 266 (p.6, “Results” section) of the manuscript as follows:

”No drop-outs or missing data were registered during the study.”

Comments 7: figure 1 pazients/ subjects with subacute post stroke  upper limb disability

Response 7: We thank the Reviewer for the observation. We have now modified the figure 1 (p.7,) of the manuscript as follows:

”119 subjects with post-stroke upper limb disability assessed for eligibility”.

Comments 8: table 1 and table 2 missing titles

Response 8: We apologize for the lack of details. We have now inserted the following titles:

·         Line 314, p.7, Table 1: “Specific telerehabilitation protocol using the end-effector robot in the Robotic Group based on patient’s clinical characteristics”;

·         Line 317, p.8, Table 2: “Demographic characteristics of study’s subjects with subacute post stroke upper limb disability divided into the two treatment groups“.

Comments 9: discussion: missing limitations of the study which is very important to state

Response 9: We understand the Reviewer’s concerns. We now have included a new section 4.1 named “Limitations” (p.12) in the manuscript. The text now reads:

Lines 433-437: “4.1. Limitations

The principal limitation of this study is the relatively small sample size. Furthermore, the absence of long-term follow-up precludes an evaluation of the sustainability of the observed improvements in both groups. Future studies should aim to address these limitations by enrolling larger sample sizes and incorporating longer follow-up periods.”
